# Topological Attention for Time Series Forecasting

**Sebastian Zeng**
Department of Computer Science
University of Salzburg
sebastian.zeng@plus.ac.at

**Florian Graf**
Department of Computer Science
University of Salzburg
florian.graf@plus.ac.at

**Christoph Hofer**
Department of Computer Science
University of Salzburg
chofer@cosy.sbg.ac.at

**Roland Kwitt**
Department of Computer Science
University of Salzburg
roland.kwitt@plus.ac.at

## Abstract

The problem of (point) forecasting *univariate* time series is considered. Most approaches, ranging from traditional statistical methods to recent learning-based techniques with neural networks, directly operate on raw time series observations. As an extension, we study whether *local topological properties*, as captured via persistent homology, can serve as a reliable signal that provides complementary information for learning to forecast. To this end, we propose *topological attention*, which allows attending to local topological features within a time horizon of historical data. Our approach easily integrates into existing end-to-end trainable forecasting models, such as `N-BEATS`, and, in combination with the latter, exhibits state-of-the-art performance on the large-scale M4 benchmark dataset of 100,000 diverse time series from different domains. Ablation experiments, as well as a comparison to a broad range of forecasting methods in a setting where only a single time series is available for training, corroborate the beneficial nature of including local topological information through an attention mechanism.

## 1 Introduction

Time series are ubiquitous in science and industry, from medical signals (e.g., EEG), motion data (e.g., speed, steps, etc.) or economic operating figures to ride/demand volumes of transportation network companies (e.g., Uber, Lyft, etc.). Despite many advances in predicting future observations from historical data via traditional statistical [5], or recent machine learning approaches [29, 42, 34, 32, 24, 44], reliable and accurate forecasting remains challenging. This is not least due to widely different and often heavily domain dependent structural properties of time related sequential data.

In this work, we focus on the problem of (point) forecasting *univariate* time series, i.e., given a length-$T$ vector of historical data, the task is to predict future observations for a given time horizon $H$. While neural network models excel in situations where a large corpus of time series is available for training, the case of only a single (possibly long) time series is equally important. The arguably most prominent benchmarks for the former type of forecasting problem are the "(M)akridakis"-competitions, such as M3 [26] or M4 [28]. While combinations (and hybrids) of statistical and machine learning approaches have largely dominated these competitions [28, see Table 4], Oreshkin et al. [29] have recently demonstrated that a pure learning-based model (`N-BEATS`) attains state-of-the-art performance. Interestingly, the latter approach is simply built from a collection of common neural network primitives which are *not* specific to sequential data.

35th Conference on Neural Information Processing Systems (NeurIPS 2021).

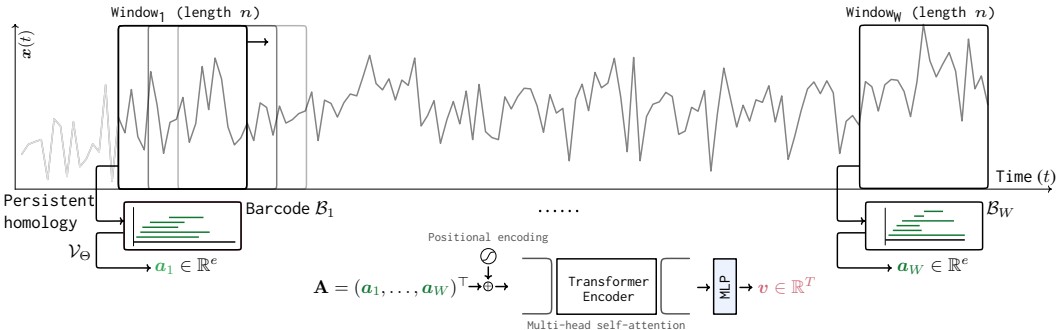

**Figure 1:** Illustration of *topological attention*, computed on time series observations $x_1, \ldots, x_T$. The signal is decomposed into a collection of $W$ overlapping windows of size $n$. For each window, a topological summary, i.e., a persistence barcode $\mathcal{B}_j$, is computed. These *local* topological summaries are then vectorized (in $\mathbb{R}^e$) via a differentiable map $\mathcal{V}_\Theta$, fed through several transformer encoder layers [41] (implementing a multi-head self-attention mechanism) with positional encoding at the input, and finally mapped to $\boldsymbol{v} \in \mathbb{R}^T$ by an MLP (best-viewed in color).

However, the majority of learning-based approaches directly operate on the raw input signal, implicitly assuming that viable representations for forecasting can be learned via common neural network primitives, composed either in a feedforward or recurrent manner. This raises the question of whether there exist structural properties of the signal, which are not easily extractable via neural network components, but offer complementary information. One prime example[1] are *topological features*, typically obtained via persistent homology [7, 13]. In fact, various approaches [31, 16, 11, 21, 15] have successfully used topological features for time series analysis, however, mostly in classification settings, for the identification of certain phenomena in dynamical systems, or for purely exploratory analysis (see Section 2).

**Contribution.** We propose an approach to incorporate *local* topological information into neural forecasting models. Contrary to previous works, we do not compute a global topological summary of historical observations, but features of short, overlapping time windows to which the forecasting model can attend to. The latter is achieved via self-attention and thereby integrates well into recent techniques, such as `N-BEATS` [29]. Notably, in our setting, computation of topological features (via persistent homology) comes with moderate computational cost, which allows application in large-scale forecasting problems.

**Problem statement.** In practice, neural forecasting models typically utilize the last $T$ observations of a time series in order to yield (point) forecasts for a given time horizon $H$. Under this perspective, the problem boils down to learning a function (parametrized as a neural network)

$$\phi : \mathbb{R}^T \to \mathbb{R}^H \quad \boldsymbol{x} \mapsto \phi(\boldsymbol{x}) = \boldsymbol{y} \ , \tag{1}$$

from a given collection of inputs (i.e., length-$T$ vectors) and targets (i.e., length-$H$ vectors). Specifically, we consider two settings, where either (1) a large collection of time series is available, as in the M4 competition, or (2) we only have access to a single time series. In the latter setting, a model has to learn from patterns *within* a time series, while the former setting allows to exploit common patterns *across* multiple time series.

## 2 Related work

**Persistent homology and time series.** Most approaches to topological time series analysis are conceptually similar, building on top of work by de Silva et al. [10] and Perea & Harer [31, 30]. Herein, time series observations are transformed into a point cloud via a *time-delay coordinate embedding* [38] from which *Vietoris-Rips (VR) persistent homology* is computed. The resulting topological summaries, i.e., persistence barcodes, are then used for downstream processing. Within this regime, Gidea et al. [15] analyze the dynamics of cryptocurrencies using persistence landscapes [6], Khasawneh et al. [21] study chatter classification in synthetic time series from turning processes

---

[1]Although learning-based approaches [36] exist to approximate topological summaries (without guarantees).

and Dłotko et al. [11] identify periodicity patterns in time series. In [22], Kim et al. actually compute one-step forecasts for Bitcoin prices and classify price patterns, essentially feeding barcode statistics as supplementary features to a MLP/CNN-based regression model. Surprisingly, very few works deviate from this pipeline, with the notable exception of [16], where VR persistent homology is *not* computed from a time-delay coordinate embedding, but rather from assembling observations (within sliding windows of size $n$) from a $d$-variate time series into a $d$-dimensional point cloud, followed by VR persistent homology computation.

Although these works clearly demonstrate that capturing the "shape" of data via persistent homology provides valuable information for time series related problems, they (1) rely on *handcrafted* features (i.e., predefined barcode summary statistics, or a fixed barcode-vectorization strategy), (2) consider topological summaries as the *single source* of information and (3) only partially touch upon forecasting problems (with the exception of [22]). Furthermore, in this existing line of work, sweeping a sliding window over the time series is, first and foremost, a way to construct a point cloud which represents the *entire* time series. Instead, in our approach, each window yields its *own* topological summary in the form of a *persistence barcode*, reminiscent to representing a sentence as a sequence of word embeddings in NLP tasks. When combined with learnable representations of persistence barcodes [19, 8], this perspective paves the way for leveraging recent techniques for handling learning problems with sequential data, such as attention [41], and allows to seamlessly integrate topological features into existing neural forecasting techniques.

**Neural network approaches to time series forecasting.** Recently, various successful neural network approaches to (mostly probabilistic) time series forecasting have emerged, ranging from auto-regressive neural networks as in DeepAR [34], to (deep) extensions of traditional state space models, such as DeepFactors [42] or DeepState [32]. While these models are inherently tailored to the sequential nature of the forecasting problem, Li et al. [24] instead rely on the concept of (log-sparse) self-attention [41], fed by the outputs of causal convolutions, and Oreshkin et al. [29] even abandon sequential neural network primitives altogether. The latter approach, solely based on operations predominantly found in feed-forward architectures, achieves state-of-the-art performance for (point) forecasts across several benchmarks, including the large-scale M4 competition.

Yet, a common factor in all aforementioned works is that raw time series observations are directly input to the model, assuming that relevant structural characteristics of the signal can be learned. While we choose an approach similar to [24], in the sense that we rely on self-attention, our work differs in that representations fed to the attention mechanism are not obtained through convolutions, but rather through a topological analysis step which, by its construction, captures the "shape" of local time series segments.

## 3 Topological attention

The key idea of topological attention is to analyze *local* segments within an input time series, $x$, through the lens of persistent homology. As mentioned in Section 2, the prevalent strategy in prior work is to first construct a point cloud from $x$ via a time-delay coordinate embedding and to subsequently compute VR persistent homology. Historically, this is motivated by studying structural properties of an underlying dynamical system, with a solid theoretical foundation, e.g., in the context of identifying periodicity patterns [31, 30]. In this regime, $x$ is encoded as a point cloud in $\mathbb{R}^n$ by considering observations within a sliding window of size $n$ as a point in $\mathbb{R}^n$.

While the time-delay embedding strategy is adequate in settings where one aims to obtain *one global* topological summary, it is inadequate when local structural properties of time series segments are of interest. Further, unless large (computationally impractical) historical time horizons are considered, one would obtain relatively sparse point clouds that, most likely, carry little information.

### 3.1 Time series as local topological summaries

Instead of a time-delay coordinate embedding, we follow an alternative strategy: a time series signal, $x$, is still decomposed into a sequence of (overlapping) windows, but not to yield a point cloud element, but rather to be analyzed in *isolation*. In the following, we only discuss the necessities specific to our approach, and refer the reader to [14, 7, 4] for a thorough treatment of persistent homology.

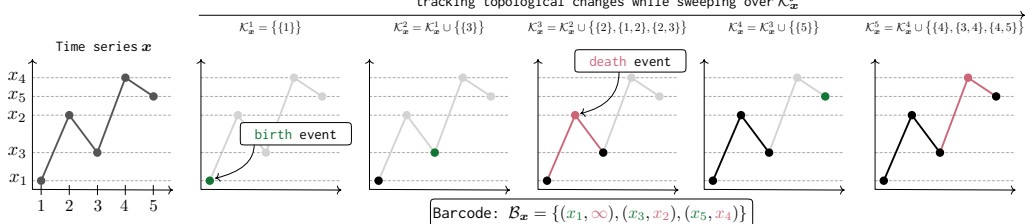

**Figure 2:** Illustration of 0-dimensional persistent homology computation for a time series $x$ of length $T = 5$. The barcode $\mathcal{B}_x$ encodes topological changes, in the form of (birth, death) tuples, as we sweep through the growing sequence $\mathcal{K}_x^1 \subseteq \cdots \subseteq \mathcal{K}_x^T$ of subsets of $\mathcal{K}$. For example, the connected component *born* at $x_3$, *dies* at $x_2$, caused by the merge with the connected component born at $x_1$ (best-viewed in color).

To topologically analyze a length-$T$ time series, over the time steps $\{1, \ldots, T\} = [T]$, in a computationally tractable manner, lets consider a 1-dimensional simplicial complex of the form

$$\mathcal{K} = \big\{\{1\}, \ldots \{T\}, \{1, 2\}, \ldots, \{T-1, T\}\big\} \ ,$$

where $\{i\}$ denote 0-simplices (i.e., vertices) and 1-simplices $\{i, j\}$ (i.e., edges) are in $\mathcal{K}$ if and only if $i$ and $j$ are two consecutive time indices. Topologically, $\mathcal{K}$ carries the connectivity properties of a time series of length $T$, which is equivalent to a straight line. This is the same for all time series of length $T$ and thus offers no discriminative information.

Persistent homology, however, lets us combine the purely topological representation of the time series with its actual values. For a specific $x$, let $a_1 \leq \cdots \leq a_T$ denote its increasingly sorted values and consider

$$\mathcal{K}_x^0 = \emptyset, \qquad \mathcal{K}_x^j = \{\sigma \in \mathcal{K} : \forall i \in \sigma : x_i \leq a_j\} \quad \text{for} \quad j \in [T] \ .$$

In fact, $\emptyset = \mathcal{K}_x^0 \subseteq \mathcal{K}_x^1 \subseteq \cdots \subseteq \mathcal{K}_x^T = \mathcal{K}$ forms an increasing sequence of subsets of $\mathcal{K}$, i.e., a *filtration*. Importantly, while $\mathcal{K}$ is the same for all time series of length $T$, the filtration, $(\mathcal{K}_x^j)_{j=0}^T$, is determined by the *values* of $x$. Persistent homology then tracks the evolution of topological features throughout this sequence and summarizes this information in the form of *persistence barcodes*.

In our specific case, as $\mathcal{K}$ is topologically equivalent to a straight line, we only get 0-dimensional features, i.e., connected components. Hence, we obtain one (0 degree) barcode $\mathcal{B}_x$. This barcode is a multiset of (birth, death) tuples, representing the birth ($b$) and death ($d$) of topological features. Informally, we may think of building $\mathcal{K}$, piece-by-piece, according to the sorting of the $x_i$, starting with the lowest value, and tracking how connected components *appear / merge*, illustrated in Fig. 2.

**Remark 3.1.** The information captured throughout this process has two noteworthy properties. First, it is *stable* in the sense that small changes in the observation values may not cause arbitrary changes in the respective barcodes, see [9]. Second, one may equally order the negative observations, i.e., $-x$, and thus obtain $\mathcal{B}_{-x}$. In that manner, the signal is analyzed from *below* and *above*.

Finally, to extract *local topological information*, we do not compute one single barcode for $x$, but one for each sliding window of size $n$, see Fig. 1. Given a decomposition of $x$ into $W$ subsequent windows, we obtain $W$ barcodes, $\mathcal{B}_1, \ldots, \mathcal{B}_W$, which constitute the entry point for any downstream operation. *Informally, those barcodes encode the evolution of local topological features over time.*

### 3.2 Barcode vectorization

Although persistence barcodes concisely encode topological features, the space of persistence barcodes, denoted as $\mathbb{B}$, carries no linear structure [40] and the nature of barcodes as multisets renders them difficult to use in learning settings. Myriad approaches have been proposed to alleviate this issue, ranging from fixed mappings into a vector space (e.g., [6, 1]), to kernel techniques (e.g., [33, 23]) and, more recently, to learnable vectorization schemes (e.g., [18, 8]). Here, we follow the latter approach, as it integrates well into the regime of neural networks. In particular, the core element in learnable vectorization schemes is a differentiable map of the form

$$\mathcal{V}_\theta : \mathbb{B} \to \mathbb{R}, \quad \mathcal{B} \mapsto \sum_{(b,d) \in \mathcal{B}} s_\theta(b, d) \ , \tag{2}$$

where $s_\theta : \mathbb{R}^2 \to \mathbb{R}$ denotes a so called *barcode coordinate function* [18], designed to preserve the *stability* property in Remark 3.1. Upon assembling a collection of $e \in \mathbb{N}$ such coordinate functions and subsuming parameters into $\Theta$, one obtains an $e$-dimensional vectorization of $\mathcal{B} \in \mathbb{B}$ via

$$\mathcal{V}_\Theta : \mathbb{B} \to \mathbb{R}^e, \quad \mathcal{B} \mapsto \mathbf{a} = \left( \mathcal{V}_{\theta_1}(\mathcal{B}), \ldots, \mathcal{V}_{\theta_e}(\mathcal{B}) \right)^\top . \tag{3}$$

Taking into account the representation of $x$ as $W$ persistence barcodes, we summarize the vectorization step as

$$\texttt{TopVec} : \mathbb{B}^W \to \mathbb{R}^{W \times e}, \quad (\mathcal{B}_1, \ldots, \mathcal{B}_W) \mapsto (\boldsymbol{a}_1, \ldots, \boldsymbol{a}_W)^\top = \left( \mathcal{V}_\Theta(\mathcal{B}_1), \ldots, \mathcal{V}_\Theta(\mathcal{B}_W) \right)^\top . \tag{4}$$

This is distinctly different to [31, 30, 21, 22] (see Section 2), where *one* barcode is obtained and this barcode is represented in a *fixed* manner, e.g., via persistence landscapes [6] or via barcode statistics.

### 3.3 Attention mechanism

In order to allow a forecasting model to attend to local topological patterns, as encoded via the $\boldsymbol{a}_j$, we propose to use the encoder part of Vaswani et al.'s [41] transformer architecture, implementing a repeated application of a (multi-head) self-attention mechanism. Allowing to attend to local time series segments is conceptually similar to Li et al. [24], but differs in the way local structural properties are captured: not via causal convolutions, but rather through the lens of persistent homology. In this setting, the scaled dot-product attention, at the heart of a transformer encoder layer, computes

$$\mathbf{O} = \mathrm{softmax}\left( \frac{(\mathbf{A}\mathbf{W}^q)(\mathbf{A}\mathbf{W}^k)^\top}{\sqrt{d_k}} \right) \mathbf{A}\mathbf{W}^v , \quad \text{with} \quad \mathbf{A} \overset{\text{Eq. (4)}}{=} (\boldsymbol{a}_1, \ldots, \boldsymbol{a}_W)^\top , \tag{5}$$

and $\mathbf{W}^q \in \mathbb{R}^{e \times d_q}, \mathbf{W}^k \in \mathbb{R}^{e \times d_k}, \mathbf{W}^v \in \mathbb{R}^{e \times d_v}$ denoting learnable (key, value, query) projection matrices. Recall that $\mathbf{A} \in \mathbb{R}^{W \times e}$ holds all $e$-dimensional vectorizations of the $W$ persistence barcodes. In its actual incarnation, one transformer encoder layer[2], denoted as $\texttt{AttnEnc} : \mathbb{R}^{W \times e} \to \mathbb{R}^{W \times e}$, computes and concatenates $M$ parallel instances of Eq. (5), (i.e., the attention heads), and internally adjusts $d_v$ such that $d = M d_v$. Composing $E$ such $\texttt{AttnEnc}$ maps, one obtains

$$
\begin{aligned}
\texttt{TransformerEncoder} &: \mathbb{R}^{W \times e} \to \mathbb{R}^{W \times e} \\
\mathbf{A} &\mapsto \texttt{AttnEnc}_1 \circ \cdots \circ \texttt{AttnEnc}_E(\mathbf{A}) .
\end{aligned}
\tag{6}
$$

Finally, we use a two-layer $\texttt{MLP} : \mathbb{R}^{We} \to \mathbb{R}^T$ (with ReLU activations) to map the vectorized output of the transformer encoder to a $T$-dimensional representation. *Topological attention* thus implements

$$
\begin{aligned}
\texttt{TopAttn} &: \mathbb{B}^W \to \mathbb{R}^T \\
(\mathcal{B}_1, \ldots, \mathcal{B}_W) &\mapsto \texttt{MLP}\big(\mathrm{vec}(\texttt{TransformerEncoder} \circ \texttt{TopVec}(\mathcal{B}_1, \ldots, \mathcal{B}_W))\big) ,
\end{aligned}
\tag{7}
$$

where $\mathrm{vec}(\cdot)$ denotes row-major vectorization operation.

**Remark 3.2.** Notably, as the domain of $\texttt{TopAttn}$ is $\mathbb{B}^W$, error backpropagation stops at the persistent homology computation. However, we remark that, given recent works on differentiating through the persistent homology computation [17, 25], one could even combine topological attention with, e.g., [24], in the sense that the outputs of causal convolutions could serve as the *filter* function for persistent homology. Error backpropagation would then consequently allow to learn this filter function.

### 3.4 Forecasting model

While the representation yielded by topological attention (see Eq. (7)) can be integrated into different neural forecasting approaches, it specifically integrates well into the $\texttt{N-BEATS}$ model of Oreshkin et al. [29]. We briefly describe the *generic* incarnation of $\texttt{N-BEATS}$ next, but remark that topological attention can be similarly integrated into the basis-expansion variant without modifications.

---

[2]omitting the additional normalization and projection layers for brevity

Essentially, the generic `N-BEATS` model is assembled from a stack of $L$ double-residual blocks. For $1 \leq l \leq L$, each block consists of a non-linear map (implemented as a MLP)

$$S^l : \mathbb{R}^T \to \mathbb{R}^h \quad , \tag{8}$$

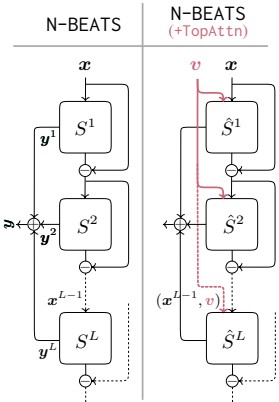

with $h$ denoting the internal dimensionality, and two subsequent maps that yield the two-fold output of the $l$-th block as

$$\boldsymbol{x}^l = \boldsymbol{x}^{l-1} - \mathbf{U}^l\big(S^l(\boldsymbol{x}^{l-1})\big), \quad \text{and} \quad \boldsymbol{y}^l = \mathbf{V}^l\big(S^l(\boldsymbol{x}^{l-1})\big) \quad ,$$

with $\mathbf{U}^l \in \mathbb{R}^{T \times h}$, $\mathbf{V}^l \in \mathbb{R}^{H \times h}$ and $\boldsymbol{x}^0 = \boldsymbol{x}$. While the $\boldsymbol{x}^l$ yield the connection to the following computation block, the $\boldsymbol{y}^l$ are used to compute the final model prediction $\boldsymbol{y} \in \mathbb{R}^H$ via (component-wise) summation, i.e., $\boldsymbol{y} = \boldsymbol{y}^1 + \cdots + \boldsymbol{y}^L$.

Importantly, in this computational chain, the $\boldsymbol{x}^l$ can be leveraged as an interface to integrate additional information. In our case, we enrich the input signal to each block by the output of the topological attention mechanism through concatenation (hence, $\hat{S}^l : \mathbb{R}^{2T} \to \mathbb{R}^h$, see figure to the top-right), i.e.,

$$\boldsymbol{x}^l_{\texttt{TopAttn}} = (\boldsymbol{x}^l, \boldsymbol{v}) \quad \text{where} \quad \boldsymbol{v} = \texttt{TopAttn}(\mathcal{B}_1, \ldots, \mathcal{B}_W) \quad . \tag{9}$$

This means that the time series signal $\boldsymbol{x}$ is (1) input (in its raw form) to `N-BEATS` and (2) its topological attention representation, $\boldsymbol{v}$, is supplied to each block as a complementary signal. In a similar manner (i.e., through concatenation), $\boldsymbol{v}$ can be included in much simpler models as well (see Section 4.2).

**Computational complexity.** Aside from the computational overhead incurred by the (multi-head) attention module, we need to compute 0-dimensional persistent homology for each sliding window. This can be done efficiently, using union find data structures, with complexity $\mathcal{O}\big(m\alpha^{-1}(m)\big)$, where $m = |\mathcal{K}| = 2n - 1$ with $n$ the sliding window size and $\alpha^{-1}(\cdot)$ denoting the inverse of the Ackermann function. As the latter grows *very* slowly, computational complexity is roughly linear for this part; see suppl. material for a detailed runtime study.

## 4 Experiments

We assess the quality of *point forecasts* in two different settings and perform ablation studies to isolate the impact of topological attention in the proposed regime.

Throughout all experiments, we compute persistent homology from $\boldsymbol{x}$ *and* $-\boldsymbol{x}$ (see Remark 3.1) using `Ripser` [3]. Barcode vectorization, see Eq. (3), is based on rational hat coordinate functions [18] with the position parameters (i.e., the locations of each coordinate function in $\mathbb{R}^2$) initialized by $k$-means++ clustering over all barcodes in the training data (with $k$ set to the number of coordinate functions). This yields a representation $\boldsymbol{a} \in \mathbb{R}^{2e}$ per sliding window. Full architecture details can be found in the suppl. material.

**Ablation setup.** When assessing each component of topological attention in isolation, we refer to `+Top` as omitting the `TransformerEncoder` part in Eq. (7), and to `+Attn` as directly feeding the time series observations to the transformer encoder, i.e., omitting `TopVec` in Eq. (7).

### 4.1 Evaluation metrics

To evaluate the quality of point forecasts, two commonly used metrics are the *symmetric mean absolute percentage error* (`sMAPE`) and the *mean absolute scaled error* (`MASE`). Letting $\hat{\boldsymbol{y}} = (\hat{x}_{T+1}, \ldots, \hat{x}_{T+H})^\top$ denote the length-$H$ forecast, $\boldsymbol{y} = (x_{T+1}, \ldots, x_{T+H})^\top$ the true observations and $\boldsymbol{x} = (x_1, \ldots, x_T)^\top$ the length-$T$ history of input observations, both scores are defined as [28]

$$\texttt{sMAPE}(\boldsymbol{y}, \hat{\boldsymbol{y}}) = \frac{200}{H} \sum_{i=1}^{H} \frac{|x_{T+i} - \hat{x}_{T+i}|}{|x_{T+i}| + |\hat{x}_{T+i}|}, \quad \texttt{MASE}(\boldsymbol{x}, \boldsymbol{y}, \hat{\boldsymbol{y}}) = \frac{1}{H} \frac{\sum_{i=1}^{H} |x_{T+i} - \hat{x}_{T+i}|}{\frac{1}{T-m} \sum_{i=m+1}^{T} |x_i - x_{i-m}|} \tag{10}$$

with $m$ depending on the observation frequency. For results on the M4 benchmark (see Section 4.3), we adhere to the competition guidelines and additionally report the *overall weighted average* (OWA) which denotes the arithmetic mean of sMAPE and MASE (with $m$ pre-specified), both measured relative to a naïve (seasonally adjusted) forecast (also provided by the M4 competition as Naive2).

## 4.2 Single time series experiments

We first consider the simple, yet frequently occurring, practical setting of *one-step* forecasts with historical observations available for only a *single* length-$N$ time series. Upon receiving a time series $\boldsymbol{x} \in \mathbb{R}^T$ (with $T \ll N$), a model should yield a forecast for the time point $T + 1$ (i.e., $H = 1$).

### 4.2.1 Dataset

To experiment with several single (but long) time series of different characteristics, we use 10 time series from the publicly available electricity [12] demand dataset[3] and four (third-party) time series of car part demands, denoted as car-parts. Based on the categorization scheme of [37], the time series are chosen such that not only *smooth* time series (regular demand occurrence and low demand quantity variation) are represented, but also *lumpy* ones (irregular demand occurrence and high demand quantity variation). For electricity, the respective percentages are 70% vs. 30%, and, for car-parts, 75% vs. 25%. All observations are *non-negative*. In case of electricity, which contain measurements in 15min intervals, we aggregate (by summation) within 7h windows, yielding a total of 3,762 observations. For car-parts, demand is measured on daily basis across a time span of 7-8 years (weekends and holidays excluded), yielding 4,644 observations on average. For each time series, 20% of held-out consecutive observations are used for testing, 5% for validation.

### 4.2.2 Forecasting model

We employ a simple incarnation of the forecasting model from Section 3.4. In particular, we replace N-BEATS by a single linear map (with bias), implementing

$$\left((\mathcal{B}_1, \ldots, \mathcal{B}_W), \boldsymbol{x}\right) \mapsto \boldsymbol{w}^\top (\boldsymbol{x}, \texttt{TopAttn}(\mathcal{B}_1, \ldots, \mathcal{B}_W)) + b \ , \tag{11}$$

with $(\boldsymbol{x}, \texttt{TopAttn}(\mathcal{B}_1, \ldots, \mathcal{B}_W))$ denoting the concatenation of the topological attention signal and the input time series $\boldsymbol{x}$, as in Eq. (9). During training, we randomly extract $T + 1$ consecutive observations from the training portion of the time series. The first $T$ observations are used as input $\boldsymbol{x}$, the observation at $T + 1$ is used as target. Forecasts for all testing observations are obtained via a length-$T$ rolling window, moved forward one step at a time.

In terms of hyperparameters for topological attention, we use a single transformer encoder layer with four attention heads and 32 barcode coordinate functions. We minimize the mean-squared-error via ADAM over 1.5k (electricity) and 2k (car-parts) iterations, respectively, with a batch size of 30. The initial learning rate of the linear map in Eq. (11) is set to 9e-2, the initial learning rates for the components of Eq. (7) are listed in Section 4.3.2, scaled up by a factor of 10. Regarding the latter, we empirically found that models trained on single time series typically require larger learning rates, most likely due to the reduced variation in the batches throughout training (as we sample from a single time series). All learning rates are annealed following a cosine learning rate schedule.

### 4.2.3 Results & Ablation study

We compare against several techniques from the literature that are readily available to a practitioner. This includes autoARIMA [20], Prophet [39], a vanilla LSTM model, as well as several approaches implemented within the GluonTS [2] library. With respect to the latter, we list results for a single-hidden-layer MLP, DeepAR [34] and MQ-CNN/MQ-RNN [43]. By Naive, we denote a baseline, yielding $x_T$ as forecast for $x_{T+1}$. *Importantly, each model is fit separately to each time series in the dataset.*

For a fair comparison, we further account for the fact that forecasting models typically differ in their sensitivity to the length of the input observations, $\boldsymbol{x}$. To this end, we cross-validate $T$ (for *all* methods) using the sMAPE on the validation set. Cross-validation points are determined by the topological attention parameters $W$ and $n$, i.e., the number and lengths of the sliding windows. For $n$ ranging from 10 to 200 and $W$ ranging from 5 to 45, we obtain a wide range of input lengths, from

---

[3]https://archive.ics.uci.edu/ml/datasets/ElectricityLoadDiagrams20112014

**Table 1:** Single time series experiments on `car-parts` and `electricity`, using the `sMAPE` as performance criterion. Listed are (1) the *average rank* (⊘ Rank) of each method, as well as (2) the *average percentual difference* (% Diff.) to the Rank-1 approach per time series. † denotes `GluonTS` [2] implementations.

**(a)** `car-parts` (#time series: 4)

| Method | ⊘ Rank | % Diff. |
|---|---|---|
| Lin.+TopAttn | **1.50** | **1.73** |
| Prophet | 2.75 | 4.05 |
| †MLP | 3.00 | 5.99 |
| †DeepAR | 3.50 | 8.24 |
| autoARIMA | 5.25 | 13.82 |
| LSTM | 6.00 | 16.54 |
| †MQ-RNN | 7.50 | 34.65 |
| Naive | 7.75 | 30.39 |
| †MQ-CNN | 7.75 | 29.49 |

**(b)** `electricity` (#time series: 10)

| Method | ⊘ Rank | % Diff. |
|---|---|---|
| Lin.+TopAttn | **1.60** | **5.43** |
| †DeepAR | 1.80 | 7.71 |
| †MLP | 2.90 | 14.54 |
| †MQ-CNN | 4.70 | 45.07 |
| autoARIMA | 5.10 | 45.31 |
| Prophet | 6.10 | 61.28 |
| LSTM | 7.40 | 69.70 |
| †MQ-RNN | 7.50 | 68.53 |
| Naive | 7.90 | 69.82 |

14 to 244. Instead of listing *absolute* performance figures, we focus on the *average rank*[4] within the cohort of methods, as well as the *average percentual difference* to the best-ranking approach per time series.

Table 1 lists the overall statistics for `electricity` and `car-parts`. We observe that, while the overall ranking per dataset differs quite significantly, `Lin.+TopAttn` consistently ranks well. Second, the average percentual difference to the best-ranking approach per time series is low, meaning that while `Lin.+TopAttn` might not yield the most accurate (wrt. the `sMAPE`) forecasts on a specific time series, it still produces forecasts of comparable quality.

Table 2 provides the same performance statistics for an ablation study of the topological attention components. Specifically, we combine the linear model of Eq. (11) with each component of topological attention in isolation.

Some observations are worth pointing out: *First*, the linear model (`Lin.`) alone already performs surprisingly well. This can possibly be explained by the fact that the task only requires one-step forecasts, for which the historical length-$T$ observations (directly preceding the forecast point) are already quite informative. *Second*, directly including topological features (i.e., `+Top`) has a confounding effect. We hypothesize that simply vectorizing local topological information from all sliding windows, *without any focus*, obfuscates relevant information, rather than providing a reliable learning signal. This also highlights the importance of attention in this context, which, even when *directly* fed with observations from each sliding window (i.e., `+Attn`), exhibits favorable performance (particularly on `car-parts`). However, in the latter strategy, the input dimensionality for the transformer encoder scales with the sliding window size $n$. Contrary to that, in case of topological attention, the

**Table 2:** Ablation study

| | ⊘ Rank | % Diff. |
|---|---|---|
| `car-parts` (#time series: 4) | | |
| Lin. | 2.75 | 0.15 |
| +Top | 3.50 | 1.45 |
| +Attn | **1.25** | 0.15 |
| +TopAttn | 2.50 | 0.64 |
| `electricity` (#time series: 10) | | |
| Lin. | 2.40 | 5.95 |
| +Top | 3.40 | 12.20 |
| +Attn | 2.30 | 2.41 |
| +TopAttn | **1.90** | 2.10 |

input dimensionality is always fixed to the of number of coordinate functions, irrespective of the sliding window size $n$.

### 4.3 Large-scale experiments on the M4 benchmark

Different to Section 4.2, we now consider having *multiple* time series of different lengths and characteristics available for training. Further, instead of one-step forecasts, the sought-for model needs to output (multi-step) point forecasts for time horizons $H > 1$.

---

[4] the `sMAPE` determines the rank of a method per time series; these ranks are then averaged over all time series

**Table 3:** Performance comparison on the M4 benchmark in terms of `sMAPE` / `OWA`, listed by subgroup. `N-BEATS` and `N-BEATS+TopAttn` denote an ensemble formed by training multiple models, varying $T$ from $2H$ to $5H$ and randomly initializing each model ten times (i.e., a total of 40 models per subgroup). Forecasts are obtained by taking the median over the point forecasts of all models. † denotes results from [27, 28].

| Method | Yearly (23k) | Quarterly (24k) | Monthly (48k) | Others (5k) | Average (100k) |
|---|---|---|---|---|---|
| †Winner M4 [35] | 13.176 / 0.778 | 9.679 / 0.847 | 12.126 / 0.836 | 4.014 /0.920 | 11.374 / 0.821 |
| †Benchmark | 14.848 / 0.867 | 10.175 / 0.890 | 13.434 / 0.920 | 4.987 / 1.039 | 12.555 / 0.898 |
| †`Naive2` | 16.342 / 1.000 | 11.011 / 1.000 | 14.427 / 1.000 | 4.754 / 1.000 | 13.564 / 1.000 |
| `N-BEATS` [29] | 13.149 / 0.776 | **9.684 / 0.845** | 12.054 / 0.829 | **3.789 / 0.857** | 11.324 / 0.814 |
| `N-BEATS+TopAttn` | **13.063 / 0.771** | 9.687 / **0.845** | **12.025 / 0.828** | 3.803 / 0.860 | **11.291 / 0.811** |

### 4.3.1 Dataset

Experiments are based on the publicly available M4 dataset[5], consisting of 100,000 time series from six domains, aggregated into six *subgroups* that are defined by the frequency of observations (i.e., yearly, quarterly, monthly, etc.). Forecasting horizons range from $H = 6$ (yearly) to $H = 48$ (hourly). The test set is fixed and contains, for all time series per subgroup, exactly $H$ observations to be predicted (starting at the last available training time point); see suppl. material for dataset statistics.

### 4.3.2 Forecasting model

We employ the forecasting model[6] of Section 3.4 and closely follow the architecture and training configuration of [29, Table 18]. Our implementation only differs in the hidden dimensionality of `N-BEATS` blocks (128 instead of 512) and in the *ensembling* step. In particular, for each forecast horizon (i.e., for each subgroup), [29] train multiple models, varying $T$ from $T = 2H$ to $T = 7H$, using ten random initializations and three separate loss functions (`sMAPE`, `MASE`, `MAPE`). One final forecast per time series is obtained by median-aggregation of each model's predictions. In our setup, we solely rely on the `sMAPE` as loss function, vary $T$ only from $T = 2H$ to $T = 5H$, but still use ten random initializations. Even with this (smaller) ensemble size (40 models per subgroup, instead of 180), `N-BEATS` alone already outperforms the winner of M4 (see Table 3). As we are primarily interested in the effect of integrating topological attention, sacrificing absolute performance for a smaller ensemble size is incidental.

In terms of hyperparameters for topological attention, the length, $n$, of sliding windows is set to $n = \lfloor 0.7 \cdot T \rfloor$, where $T$ varies per subgroup as specified above. The model uses 20 transformer encoder layers with four attention heads and 64 structure elements for barcode vectorization. For optimization, we use ADAM with initial learning rates of 1e-3 (for `N-BEATS` and the `MLP` part of Eq. (7)), 8e-3 (`TopVec`) and 5e-3 (`TransformerEncoder`). All learning rates are annealed according to a cosine learning rate schedule over 5,000 iterations with a batch size of 1,024.

### 4.3.3 Results & Ablation study

Table 3 lists the `sMAPE` and `OWA` for the winner of the M4 competition [35], as well as the `Naive2` baseline (with respect to which the `OWA` is computed) and the M4 benchmark approach, obtained as the arithmetic mean over simple, Holt, and damped exponential smoothing.

In terms of the `OWA`, we see an overall 0.4% improvement over `N-BEATS` and a 1.2% improvement over the M4 winner [35]. In particular, topological attention performs well on the large yearly / monthly subgroups of 23k and 48k time series, respectively. While `OWA` scores are admittedly quite close, the differences are non-negligible, considering the large corpus of 100k time series. In fact, several methods in the official M4 ranking differ by an even smaller amount with respect to the `OWA` measure.

**Table 4:** Ablation study

| Method | sMAPE | OWA |
|---|---|---|
| `N-BEATS` | 11.488 | 0.827 |
| +Top | 11.505 | 0.920 |
| +Attn | 11.492 | 0.826 |
| +TopAttn | **11.466** | **0.824** |

---

[5]available at https://github.com/Mcompetitions/M4-methods
[6]based on the `N-BEATS` reference implementation https://github.com/ElementAI/N-BEATS

Similar to the ablation results of Section 4.2, the ablation study in Table 4 (conducted for $T = 2H$ only) reveals the beneficial effect of topological attention, in particular, the beneficial nature of allowing to *attend* to local topological features. Contrary to the ablation in Table 2, we see that in this large-scale setting, *neither* topological features (`+Top`), *nor* attention (`+Attn`) *alone* yield any improvements over the already strong `N-BEATS` model. In fact, when integrated separately into `N-BEATS`, both components even deteriorate performance in terms of the `sMAPE`.

**Number of parameters.** Regarding a comparison of `N-BEATS` and `N-BEATS+TopAttn` in terms of the *number of parameters*, we note that the particular `N-BEATS` incarnation in our experiments has a total of ≈1.7M parameters; our approach adds about 700k additional parameters. To assess whether the reported improvements are simply due to a larger overall model, we ran additional experiments increasing the size of `N-BEATS` (by doubling the hidden dimensionality) to ≈6.4M parameters. As in Table 4, this study was performed using a historical time horizon of $T = 2H$ and all results are averaged over 10 random initializations of each model. In terms of the `OWA`, the *larger* `N-BEATS` model achieves an average score of 0.826 *vs.* 0.827 for `N-BEATS` and 0.824 for `N-BEATS+TopAttn` (the latter two numbers correspond to the results in Table 4). Hence, substantially increasing the number of parameters for `N-BEATS` indeed improves the overall score (as expected), but does not reach the `OWA` result of our model.

## 5  Conclusion

While several prior forecasting works have pointed out the relevance of local structural information within historical observations (e.g., [24]), it is typically left to the model to learn such features from data. Instead, we present a direct approach for capturing the "shape" of local time series segments via persistent homology. Different to the typical application of the latter in signal analysis, we capture the evolution of topological features over time, rather than a global summary, and allow a forecasting model to attend to these local features. The so obtained *topological attention mechanism* yields a complementary representation that easily integrates into neural forecasting approaches. In combination with `N-BEATS` [29], for instance, large-scale experiments on the M4 benchmark provide evidence that including topological attention indeed allows to obtain more accurate point forecasts.

**Societal impact.** Due to the ubiquity of time series data, forecasting in general, certainly touches upon a variety of societally relevant and presumably sensible areas. As our work has potential impact in that sense, we perform large-scale experiments over a wide variety of time series from different domains, thereby obtaining a broad picture of the overall forecasting performance.

Source code is publicly available at https://github.com/plus-rkwitt/TAN.

## Acknowledgments & Disclosure of funding

This research was supported in part by the Austrian Science Fund (FWF): project FWF P31799-N38 and the Land Salzburg (WISS 2025) under project numbers 20102- F1901166-KZP and 20204-WISS/225/197-2019. The first author also gratefully acknowledges financial support from Porsche Holding Austria and Land Salzburg within the WISS 2025 project "KFZ" (P1900123). Finally, we like to thank all anonymous reviewers for their constructive feedback during the review process.

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
