# Supplementary material

## A  Evaluation criteria

In the following, we first replicate the definition of two commonly used, scale-free evaluation metrics, i.e., the *symmetric mean absolute percentage error* (sMAPE) and the *mean absolute scaled error* (MASE), see **[Manuscript, Section 4.1]**. These scale-free metrics are standard in the practice of forecasting and used across all experiments in the manuscript. Subsequently, we define the *overall weighted average* (OWA), i.e., a M4 competition specific performance measure used to rank competition entries. We also provide a toy calculation example.

Letting $\hat{\boldsymbol{y}} = (\hat{x}_{T+1}, \ldots, \hat{x}_{T+H})^\top$ denote the length-$H$ forecast, $\boldsymbol{y} = (x_{T+1}, \ldots, x_{T+H})^\top$ the true observations and $\boldsymbol{x} = (x_1, \ldots, x_T)^\top$ the length-$T$ history of input observations, both metrics are defined as [28]

$$\text{sMAPE}(\boldsymbol{y}, \hat{\boldsymbol{y}}) = \frac{200}{H} \sum_{i=1}^{H} \frac{|x_{T+i} - \hat{x}_{T+i}|}{|x_{T+i}| + |\hat{x}_{T+i}|}, \ \text{MASE}(\boldsymbol{x}, \boldsymbol{y}, \hat{\boldsymbol{y}}) = \frac{1}{H} \frac{\sum_{i=1}^{H} |x_{T+i} - \hat{x}_{T+i}|}{\frac{1}{T-m} \sum_{i=m+1}^{T} |x_i - x_{i-m}|} \quad (12)$$

with $m$ depending on the observation frequency. In the M4 competition [28], the frequencies per subgroup are: 12 for monthly, four for quarterly, 24 for hourly and one for yearly / weekly / daily data. To obtain the OWA of a given forecast method, say Forecast, we compute [28]

$$\text{OWA}_{\text{Forecast}} = \frac{1}{2} \left[ \frac{\text{sMAPE}_{\text{Forecast}}}{\text{sMAPE}_{\text{Naive2}}} + \frac{\text{MASE}_{\text{Forecast}}}{\text{MASE}_{\text{Naive2}}} \right] \quad . \quad (13)$$

Thus, if Forecast displays a MASE of 1.63 and a sMAPE of 12.65% across the 100k time series of M4, while Naive2 displays a MASE of 1.91 and a sMAPE of 13.56%, the relative MASE and sMAPE of Forecast would be 1.63/1.91 = 0.85 and 12.65/13.56 = 0.93, respectively, resulting in an OWA of $(0.93 + 0.85)/2 = 0.89$. According to [28], this indicates that, on average, Forecast is about 11% more accurate than Naive2, taking into account both sMAPE and MASE.

**Performance criteria for single time series experiments.** In our single time series experiments of **[Manuscript, Section 4.2]**, we use the sMAPE as an underlying evaluation measure and compute the following two statistics: first, the average rank ($\oslash$ **Rank**) of each method based on the rank on each single time series in car-parts and electricity (both datasets are treated separately); and, second, the average percentual difference (% **Diff**) to the best approach per time series. A calculation example, for four *hypothetical* models and three time series ($\text{TS}_0$, $\text{TS}_1$, $\text{TS}_2$), is listed in Table 5.

**Table 5:** Example calculation of the *average rank* ($\oslash$ **Rank**) and the *average percentual difference* (% **Diff**), as reported in [Manuscript, Table 1]. In this calculation example, lower scores are better. For instance, on $\text{TS}_1$, the rank of Model C is 3 and the percentual difference to the best performing model on $\text{TS}_1$ (i.e., Model A) is $(1.0 - 11.8/12.2) \times 100 = 3.28$.

|         | $\text{TS}_0$ | $\text{TS}_1$ | $\text{TS}_2$ | $\oslash$ **Rank** | % **Diff** |
|---------|---------------|---------------|---------------|--------------------|------------|
| Model A | 14.4 (3, 9.66) | 11.8 (1, 0.00) | 10.5 (2, 3.81) | **2.00** | **4.49** |
| Model B | 14.3 (2, 8.39) | 12.1 (2, 2.48) | 10.8 (3, 6.48) | **2.33** | **5.78** |
| Model C | 13.1 (1, 0.00) | 12.2 (3, 3.28) | 11.1 (4, 9.01) | **2.67** | **4.09** |
| Model D | 14.5 (4, 9.66) | 13.1 (4, 9.92) | 10.1 (1, 0.00) | **3.00** | **6.52** |

# B   Dataset details

For completeness, Table 6 replicates [29, Table 2], providing an overview of the key statistics for the M4 competition dataset. For all results listed in the main manuscript, the subgroups *Weekly*, *Daily* and *Hourly* are aggregated into **Others**, accounting for 5,000 time series overall.

**Table 6:** Description / Statistics for the M4 competition dataset.

| Type | Frequency / Horizon | | | | | | Total |
|------|------------|-------------|-------------|------------|-----------|------------|---------|
|      | Yearly / 6 | Quarterly / 8 | Monthly / 18 | Weekly / 13 | Daily / 14 | Hourly / 48 | |
| Demographic | 1,088 | 1,858 | 5,728 | 24 | 10 | 0 | 8,708 |
| Finance | 6,519 | 5,305 | 10,987 | 164 | 1,559 | 0 | 24,534 |
| Industry | 3,716 | 4,637 | 10,017 | 6 | 422 | 0 | 18,798 |
| Macro | 3,903 | 5,315 | 10,016 | 41 | 127 | 0 | 19,402 |
| Micro | 6,538 | 6,020 | 10,975 | 112 | 1,476 | 0 | 25,121 |
| Other | 1,236 | 865 | 277 | 12 | 633 | 414 | 3,437 |
| **Total** | 23,000 | 24,000 | 48,000 | 359 | 4,227 | 414 | 100,000 |
| Min. Length | 19 | 24 | 60 | 93 | 107 | 748 | |
| Max. Length | 841 | 874 | 2812 | 2610 | 9933 | 1008 | |
| Mean Length | 37.3 | 100.2 | 234.3 | 1035.0 | 2371.4 | 901.9 | |
| SD Length | 24.5 | 51.1 | 137.4 | 707.1 | 1756.6 | 127.9 | |
| % Smooth | 82% | 89% | 94% | 84% | 98% | 83% | |
| % Erratic | 18% | 11% | 6% | 16% | 2% | 17% | |

Table 7 lists key statistics for the `car-parts` and the `electricity` time series we use in **[Manuscript, Section 4.2]**. Notably, there are no time series categorized into the *erratic* category, according to Syntetos et al. [37]. As `car-parts` is proprietary, Fig. 3 additionally shows a visualization of all observations from the four spare part demand time series.

**Table 7:** Description / Statistics for the `car-parts` and `electricity` time series.

| | car-parts | electricity |
|------|-----------|-------------|
| Frequency / Horizon | Daily / 1 | 7Hourly / 1 |
| Total | 4 | $10^7$ |
| Min. Length | 4507 | 3762 |
| Max. Length | 4783 | 3762 |
| Mean Length | 3644 | 3762 |
| SD Length | 137 | 0 |
| % Smooth | 75% | 70% |
| % Lumpy | 25% | 30% |

---

[7]In reference to [12], time series IDs are: MT_$i$ for $i \in \{14, 127, 130, 183, 238, 271, 318, 332, 333, 353\}$.

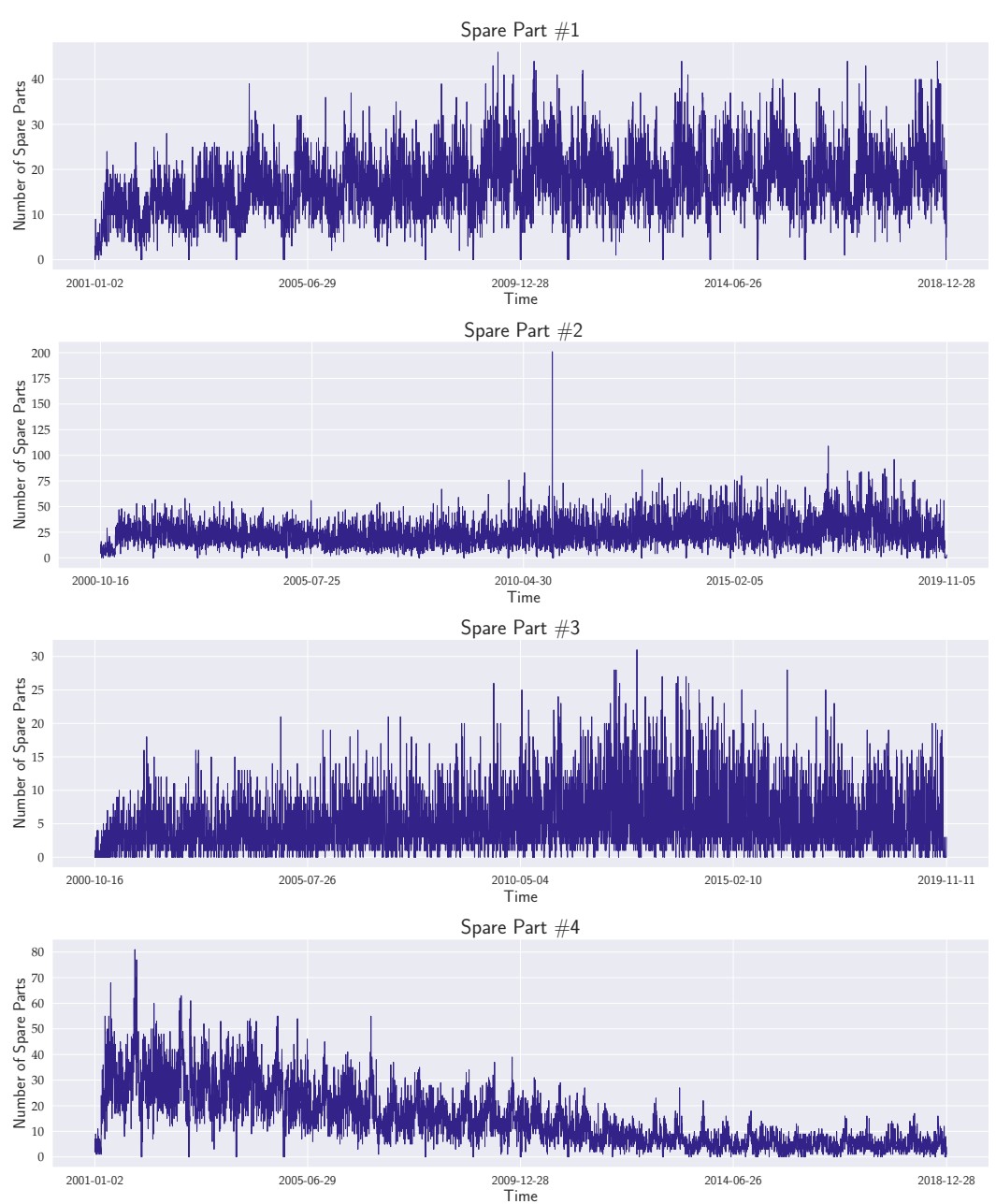

**Figure 3:** Visualization of the four proprietary `car-parts` time series.

## C Additional results

Table 8 replicates [**Manuscript, Table 3**], listing sMAPE / OWA statistics on M4, as well as the corresponding MASE / OWA statistics[8]. Table 9 lists results, split by time series **domains**.

**Table 8:** Performance comparison on the M4 benchmark in terms of (a) sMAPE / OWA and (b) MASE / OWA, listed by subgroup. N-BEATS and N-BEATS+TopAttn denote an ensemble formed by training multiple models, varying $T$ from $2H$ to $5H$ and randomly initializing each model ten times (i.e., a total of 40 models per subgroup). Forecasts are obtained by taking the median over the point forecasts of all models. † denotes results from [27, 28].

**(a)** sMAPE / OWA

| Granularity | Total | †Winner M4 | †Benchmark | †Naive2 | N-BEATS [29] | N-BEATS+TopAttn |
|---|---|---|---|---|---|---|
| Yearly | (23k) | 13.176 / 0.778 | 14.848 / 0.867 | 16.342 / 1.000 | 13.149 / 0.776 | **13.063 / 0.771** |
| Quarterly | (24k) | 9.679 / 0.847 | 10.175 / 0.890 | 11.011 / 1.000 | **9.684** / 0.845 | 9.687 / **0.845** |
| Monthly | (48k) | 12.126 / 0.836 | 13.434 / 0.920 | 14.427 / 1.000 | 12.054 / 0.829 | **12.025 / 0.828** |
| Weekly | (359) | 7.817 / 0.851 | 8.944 / 0.926 | 9.191 / 1.000 | 6.447 / 0.703 | **6.361 / 0.699** |
| Daily | (4,227) | 3.170 / 1.046 | 2.980 / 0.978 | 3.045 / 1.000 | **2.976 / 0.974** | 2.979 / 0.975 |
| Hourly | (414) | **9.328 / 0.440** | 22.053 / 1.556 | 18.383 / 1.000 | 10.040 / 0.464 | 10.271 / 0.483 |
| **Average** | (100k) | 11.374 / 0.821 | 12.555 / 0.898 | 13.564 / 1.000 | 11.324 / 0.814 | **11.291 / 0.811** |

**(b)** MASE / OWA

| Granularity | Total | †Winner M4 | †Benchmark | †Naive2 | N-BEATS [29] | N-BEATS+TopAttn |
|---|---|---|---|---|---|---|
| Yearly | (23k) | 2.980 / 0.778 | 3.280 / 0.867 | 3.974 / 1.000 | 2.972 / 0.776 | **2.950 / 0.771** |
| Quarterly | (24k) | 1.118 / 0.847 | 1.173 / 0.890 | 1.371 / 1.000 | **1.111 / 0.845** | 1.112 / **0.845** |
| Monthly | (48k) | 0.884 / 0.836 | 0.966 / 0.920 | 1.063 / 1.000 | 0.875 / 0.829 | **0.874 / 0.828** |
| Weekly | (359) | 2.356 / 0.851 | 2.432 / 0.926 | 2.777 / 1.000 | **1.950** / 0.703 | 1.953 / **0.699** |
| Daily | (4,227) | 3.446 / 1.046 | 3.203 / 0.978 | 3.278 / 1.000 | **3.183 / 0.974** | 3.188 / 0.975 |
| Hourly | (414) | **0.893 / 0.440** | 4.582 / 1.556 | 2.395 / 1.000 | 0.917 / 0.464 | 0.974 / 0.483 |
| **Average** | (100k) | 1.536 / 0.821 | 1.663 / 0.898 | 1.912 / 1.000 | 1.516 / 0.814 | **1.511 / 0.811** |

**Table 9:** Performance comparison on the M4 benchmark in terms of (a) sMAPE$_{\text{N-BEATS}}$ / sMAPE$_{\text{N-BEATS+TopAttn}}$ and (b) MASE$_{\text{N-BEATS}}$ / MASE$_{\text{N-BEATS+TopAttn}}$, listed by subgroup and **domain**. N-BEATS and N-BEATS+TopAttn denote the same models as in Table 8.

**(a)** sMAPE$_{\text{N-BEATS}}$ / sMAPE$_{\text{N-BEATS+TopAttn}}$

| Granularity | Demographic (8,7k) | Finance (24,5k) | Industry (18,8k) | Macro (19,4k) | Micro (25,1k) | Other (3,5k) |
|---|---|---|---|---|---|---|
| Yearly | 9.640 / 9.694 | 14.029 / 13.879 | 16.645 / 16.523 | 13.450 / 13.400 | 10.700 / 10.654 | 13.094 / 13.000 |
| Quarterly | 9.908 / 9.933 | 11.158 / 11.161 | 8.822 / 8.832 | 9.182 / 9.178 | 9.919 / 9.922 | 6.222 / 6.173 |
| Monthly | 4.605 / 4.599 | 13.629 / 13.625 | 12.918 / 12.913 | 12.490 / 12.428 | 13.180 / 13.122 | 11.987 / 11.932 |
| Weekly | 1.401 / 1.403 | 7.598 / 7.516 | 2.563 / 2.548 | 11.303 / 10.837 | 3.658 / 3.681 | 12.204 / 12.112 |
| Daily | 6.300 / 6.313 | 3.442 / 3.446 | 3.831 / 3.832 | 2.532 / 2.532 | 2.288 / 2.291 | 2.901 / 2.901 |
| Hourly | | | | | | 9.787 / 9.997 |
| **Average** | **6.358** / 6.367 | 12.513 / **12.472** | 12.437 / **12.413** | 11.709 / **11.665** | 11.070 / **11.034** | 8.997 / **8.971** |

**(b)** MASE$_{\text{N-BEATS}}$ / MASE$_{\text{N-BEATS+TopAttn}}$

| Granularity | Demographic (8,7k) | Finance (24,5k) | Industry (18,8k) | Macro (19,4k) | Micro (25,1k) | Other (3,5k) |
|---|---|---|---|---|---|---|
| Yearly | 2.410 / 2.428 | 3.086 / 3.055 | 3.021 / 2.996 | 2.956 / 2.932 | 2.994 / 2.981 | 2.647 / 2.616 |
| Quarterly | 1.234 / 1.238 | 1.110 / 1.111 | 1.075 / 1.077 | 1.123 / 1.121 | 1.128 / 1.128 | 0.866 / 0.861 |
| Monthly | 0.864 / 0.862 | 0.912 / 0.912 | 0.936 / 0.935 | 0.878 / 0.876 | 0.790 / 0.788 | 0.780 / 0.778 |
| Weekly | 1.782 / 1.839 | 1.661 / 1.634 | 3.724 / 3.808 | 2.042 / 2.114 | 2.393 / 2.399 | 0.910 / 0.899 |
| Daily | 9.604 / 9.641 | 3.396 / 3.402 | 3.784 / 3.787 | 3.198 / 3.205 | 2.597 / 2.603 | 3.519 / 3.520 |
| Hourly | | | | | | 0.903 / 0.971 |
| **Average** | **1.148** / 1.151 | 1.695 / **1.687** | 1.447 / **1.443** | 1.380 / **1.374** | 1.558 / **1.554** | 1.993 / **1.989** |

---

[8]Detailed results for *Weekly*, *Daily* and *Hourly* are not listed in [27, 28], but available here.

# D Hyperparameter settings

Hyperparameter settings for our single time series experiments of [**Manuscript, Section 4.2**] and the large-scale M4 experiments of [**Manuscript, Section 4.3**] are listed in Tables 10 and 11.

**Table 10:** *Single time series experiment* hyperparameters for `car-parts` and `electricity` data.

| Parameters | car-parts
Daily | electricity
7Hourly |
|---|---|---|
| Iterations | 2k | 1.5k |
| Loss | MSE | |
| $H$ (Forecast horizon) | 1 | |
| Lookback period(s), $T$ | $14H$ - $244H$ | |
| Batch size | 30 | |
| Attention heads | 4 | |
| Barcode coordinate functions | 32 | |
| Encoder-layers | 1 | |
| Hidden dimension | 128 | |

As mentioned in the manuscript, for M4 experiments with `N-BEATS` (and `N-BEATS+TopAttn`), we closely follow the *generic* `N-BEATS` parameter configuration of Oreshkin et al. [29, Table 18]; any additional parameters (for our `N-BEATS+TopAttn` approach) are highlighted in red. Note that we also mark *Hidden dimension* in red, as this is not only the hidden dimension of the `N-BEATS` blocks ("width" in [29, Table 18]), but we equally use this setting for the hidden dimension of the transformer encoder layers.

**Table 11:** *Large-scale experiment* hyperparameters across all subsets of the M4 dataset. Parameters specific to `N-BEATS+TopAttn` are highlighted in red. For a detailed description of the `N-BEATS` parameters, we refer to [29, Section D.1].

| Parameters | M4 | | | | | |
| | Yearly | Quarterly | Monthly | Weekly | Daily | Hourly |
|---|---|---|---|---|---|---|
| $H$ (Forecast horizon) | 6 | 8 | 18 | 13 | 14 | 48 |
| $L_H$ | 1.5 | 1.5 | 1.5 | 10 | 10 | 10 |
| Iterations | 5k | | | | | |
| Loss | sMAPE | | | | | |
| Lookback period(s), $T$ | $2H, 3H, 4H, 5H$ | | | | | |
| Batch size | 1024 | | | | | |
| Attention heads | 4 | | | | | |
| Barcode coordinate functions | 64 | | | | | |
| Encoder-layers | 20 | | | | | |
| Hidden dimension | 128 | | | | | |
| Double-residual blocks | 1 | | | | | |
| Block-layers | 4 | | | | | |
| Stacks | 30 | | | | | |

# E    Sliding window configurations

Lets assume we have, at one point in training, a randomly extracted training portion of $T + H$ consecutive observations from a length-$N$ time series ($T \ll N$). We use the first $T$ observations as (1) *raw* input signal $\boldsymbol{x}$ to our models and (2) for extraction of complementary *local topological properties*. The $H$ consecutive observations, starting at $T + 1$, are used as target (to compute the mean-squared-error, or the `sMAPE` for instance).

*Throughout all experiments, sliding windows are moved forward by one observation a time.*

For extracting local topological properties from $\boldsymbol{x}$ (of length $T$) via persistent homology, two parameters are necessary: the parameter $W$ determines the number of overlapping sliding windows and the parameter $n$ determines the length of a single sliding window (i.e., $n$ observations). For each sliding window, we obtain one barcode (or two, if $-\boldsymbol{x}$ is taken into account).

**Single time series experiments [Manuscript, Section 4.2].** In this setting, $H = 1$, as we compute one-step forecasts. Since, typically, forecast models differ in their sensitivity to the length $T$ of the input observations $\boldsymbol{x}$, we cross-validate $T$ (for all methods) using the `sMAPE` on the validation set.

The collection of different $T$'s used for cross-validation is constructed based on the following consideration: first, for persistent homology computation, we need a reasonable amount, $n$, of observations in each sliding window; and, second, we need a reasonable amount, $W$, of sliding windows for self-attention. Hence, we choose (1) $W \geq 10$ and (2) $n \leq 45$. As the sliding windows move forward by one observation at a time, for one specific choice of $(W, n)$, we get $T = W + n - 1$. Varying $W \in \{5, 25, 45\}$ and $n \in \{10, 20, 50, 70, 100, 150, 200, 232\}$ thus determines the length, $T$, of the input vector $\boldsymbol{x}$. For instance, setting $(W, n) = (5, 10)$ gives a decomposition of $\boldsymbol{x}$ (of length 14), into 5 subsequent windows of length 10 for which persistent homology is computed. Overall, in the described setup, $T$ ranges from 14 to 244.

**Large-scale experiments [Manuscript, Section 4.3].** In this setting, $H > 1$. For comparability with `N-BEATS`, we stick to the original setup of considering input lengths as multiples of the forecast horizon (which is specific to each subgroup in M4). In particular, $T$ ranges from $2H$ to $5H$, see Table 11. As an example, on M4 *Yearly*, this yields a range of $T$ from 12 to 30. As mentioned in **[Manuscript, Section 4.3.2]**, we set the sliding window length $n = \lfloor 0.7 \cdot T \rfloor$ and the number, $W$, of such windows is thus determined by $(T, n)$.

# F Ensemble size

As described in **[Manuscript, Section 4.3.2]**, we ensemble 40 models to obtain forecasts for each subgroup of the M4 dataset. One ensemble is formed per subgroup and consists of training N-BEATS, or N-BEATS+TopAttn, respectively, with 10 random initializations for four different values of $T$, i.e., $2H, 3H, 4H, 5H$ (where $H$ denotes the specific forecast horizon prescribed per subgroup), using the sMAPE as a loss function. In case of *Yearly* for instance, $H = 6$, see Table 6. Fig. 4 shows a comparison of N-BEATS and N-BEATS+TopAttn over the ensemble size, illustrating that N-BEATS+TopAttn equally benefits from a larger ensemble.

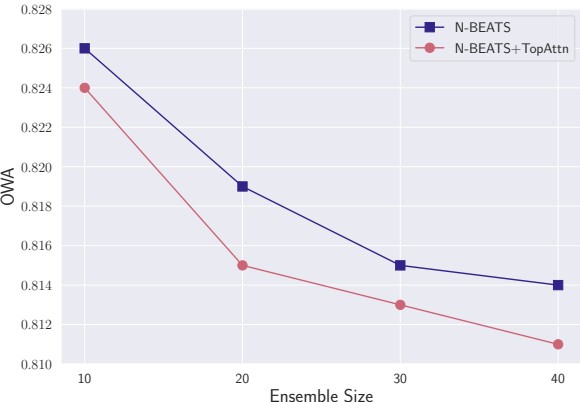

**Figure 4:** Comparison of N-BEATS and N-BEATS+TopAttn in terms of varying the ensemble size. At the maximum ensemble size of 40, the OWA corresponds to the OWA reported in, e.g., Table 8. In the figure, the values on the $x$-axis correspond to ensembles formed from all random initializations of models trained with historical time horizons up to $2H, 3H, 4H, 5H$.

Notably, in [29] the ensemble is larger, as, in addition to training models with the sMAPE as loss, the MAPE and MASE are used as well, and $T$ scales up to $7H$, resulting in 180 models in total. To rule out diminishing effects when further increasing the ensemble size, specifically by models trained with losses other than the sMAPE, we added 20 more models trained with the MASE for time horizons $2H$ and $3H$. This gives an overall ensemble size of 60 (i.e., two more horizons and 10 random initializations each).

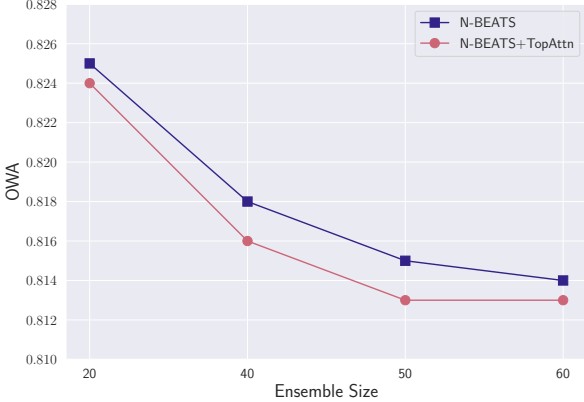

**Figure 5:** see Fig. 4, but with more models added to the ensemble, specifically 20 models trained using the MASE as a training criterion for historical time horizons $2H$ and $3H$.

# G Model details

In this section, we describe the details for the models used in the *single time series* experiments of [**Manuscript, Section 4.2**].

**Prophet.** We use the publicly available Python implementation of `Prophet`[9] with *default* parameter choices.

**autoARIMA.** We use the publicly available Python implementation of `autoARIMA`[10]. In terms of hyperparameters, the initial number of time lags of the auto-regressive ("AR") and the moving-average ("MA") model is set to 1, bounded by its maximum 6. The period for seasonal differencing is equal to 5; the order of first-differencing and of seasonal differencing is set to 2 and 0, respectively.

**LSTM.** We implement a `LSTM` model with hidden dimensionality 128, 8 recurrent layers and a dropout layer on the outputs of each LSTM layer with dropout probability of 0.3. Outputs of the LSTM are fed to a subsequent single-hidden-layer `MLP` with hidden dimensionality equal to 64, including batch normalization and ReLU activation. Initial learning rate and weight decay are set to 1e-3 and 1.25e-5, respectively. We minimize the mean-squared-error (MSE) via ADAM over 1.5k (`electricity`) and 2k (`car-parts`) iterations, respectively, using a batch size of 30. All learning rates are annealed following a cosine learning rate schedule.

For **DeepAR**, **MQ-CNN**, **MQ-RNN** and the **MLP** baseline, we use the publicly available `GluonTS` [2] implementations[11], mostly with *default* parameter choices. We only adjust the number of (maximum) training epochs to 20 (for comparability to our approach, where we count iterations), change the hidden dimensionality of the **MLP** to 64 and set the batch size to 30.

---

[9] https://facebook.github.io/prophet/
[10] https://alkaline-ml.com/pmdarima/
[11] https://ts.gluon.ai

## H   System setup

All experiments were executed on an Ubuntu Linux 20.04 system, using `PyTorch` v1.7.0 (CUDA 10.1), 128 GB of RAM and 16 Intel(R) Core(TM) i9-10980XE CPUs.

## I   Persistent homology – Runtime study

To back up the "near-linear runtime" statement for 0-dimensional persistent homology computation in the proposed regime (see **[Manuscript, Section 3.4]**), Fig. 6 shows a runtime plot (using `Ripser`[12]) over 10,000 sliding window sizes, $n$, in the range $[5, 2000]$. The system setup for these runtime experiments is given in Section H. Fig. 6 clearly corroborates the statement from the manuscript.

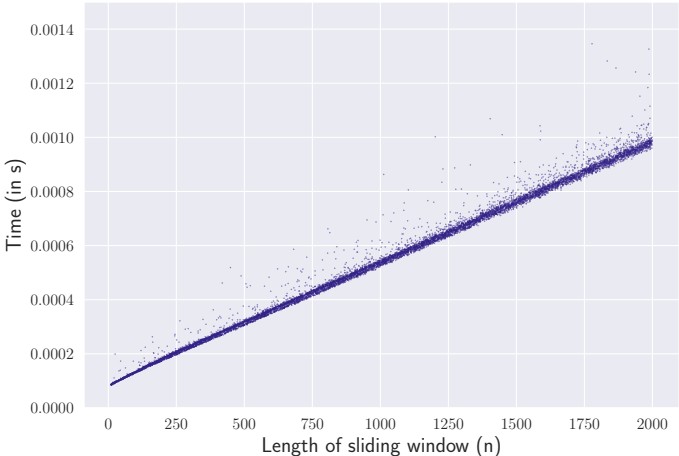

**Figure 6:** Runtime (in seconds) for 0-dimensional persistent homology computation from observations within a sliding window, varying in length, $n$, from $[5, 2000]$.

---

[12]https://github.com/Ripser/ripser