# OpenReview forum: "Topological Attention for Time Series Forecasting"
_NeurIPS.cc/2021/Conference — NeurIPS 2021 Poster_

### Official Review · Reviewer_RPLH · 2021-07-05

**Rating:** 6
**Confidence:** 4

**Summary:**

This paper proposes a new approach for one-dimensional time-series forecasting that is based on augmenting the data with topological information. Specifically, the authors design an attention module that extract the local connections in their windowed persistent homology barcodes, and concatenates this information to the original time-series. The authors evaluate their method on two scenarios: a single series and multiple time-series elements. Their techniques shows modest improvement over existing approaches.

**Limitations And Societal Impact:**

See above.

**Main Review:**

This submission proposes an interesting idea: in addition to operating on the raw data, it suggests to augment the data with local topological information that is drawn from persistent homology computations of the data. In practice, the proposed method includes a module that generates the persistent barcodes, and an attention module which extract local connections in the windowed barcodes. To the best of my knowledge, this idea is novel and was not suggested before. In general, I like this approach, as it allows to connect a fundamental problem (time-series forecasting) with an established theory and practice (persistent homology). As such, this work has the potential to inspire further research that combines persistent homology in the context of time-series processing or other machine-learning problems. Overall, the exposition is clear, and the authors made an effort to include many details about persistent barcodes and their computation. I believe their description in 3.1 & 3.2 could be further, however, I do not have a concrete suggestion on how to do it---perhaps use less general descriptions. For instance, you can directly write what is the particular $mathcal{V}_\theta$ you used right below eq. 2.

The main issue I found with this submission is the somewhat on-par results in the multi-series setting. In particular, the improvement over N-BEATS in Table 3 seems marginal. Moreover, the comment at the end of pg. 8 makes me think that the problem is more fundamental. Namely, since the authors did not use the ``best'' version of N-BEATS, and improve on a subpar version of it, it might be that the proposed mechanism yields no improvement when added to the full N-BEATS. At this point---we have no way of knowing. With that being said, I am still on the positive side, given the potential of this work.


**Time Spent Reviewing:**

4

---

> ### Author Response · Authors · 2021-08-09
> **Response to Questions/Comments**
>
> &#9654; **Improved exposition in Section 3**
>
> We agree that Sections 3.1 & 3.2 could be more specific. We deliberately kept them as general as possible, as especially the vectorization part, can be easily switched out. As suggested, we will include a precise description of the specific vectorization strategy ($\mathcal{V}_{\theta}$) in Section 3.2.
>
>  Regarding Section 3.1, our main objective was to balance a mathematically precise
>  description without loosing track of the key idea for computing persistent homology
>  in our setting. We think, supplementary to Figure 2, a short code snipped on how to actually compute the barcode will be helpful and we will provide this in the appendix.
>
>  &#9654; **Regarding on-par results with NBEATS**
>
> The reviewer is correct in that the relative improvement over vanilla NBEATS is rather small. Yet, the first 5 top-ranking results on M4 lie in a range of 1%-4.5% difference to the winner of the M4 competition. In fact, even a ~1% improvement (e.g., 2nd-best over 3rd-best or 3rd-best over 4th-best on M4, see [[28]](https://www.sciencedirect.com/science/article/pii/S0169207019301128)) is notable, considering the massive amount of 100k time series. Our margin to NBEATS also resides in that range.
>
>  &#9654; **On the use of a sub-par version of NBEATS**
>
>  We like to clarify that we *did not* limit the NBEATS model itself, we *only* constrained the ensemble size. Specifically, the ensemble differs in that (1) we only train with SMAPE and did not include models trained with MASE and MAPE and (2) we did not train models with historical windows of size 6H and 7H. Overall, this yields an ensemble size of 40, compared 180 for *full NBEATS*. Across all tested ensemble sizes (10,20,30,40), topological attention yielded consistent improvements, see Figure F.1.
>
>  Yet, we do agree with the reviewer, that the benefit of adding topological attention to a *larger* ensemble should be explored. Given the current time frame for these initial comments, we managed to run several *additional experiments* using MASE as training criterion. This adds 10 x 2 additional models to the ensemble (i.e., two horizons, 2H/3H, times 10 random initializations) and enlarges our ensemble size plot of Figure F.1 to 60 models. As in the original Figure F.1, NBEATS+TopAttn **always** remains below vanilla NBEATS in terms of OWA for all additional ensemble sizes (i.e., 50,60) with a consistent gap (of ~0.002 in OWA).
>
> We will complete the ensemble with models trained with MAPE and the remaining horizons and update the ensemble plot in a final version.

---

### Official Review · Reviewer_Erx9 · 2021-07-12

**Rating:** 6
**Confidence:** 4

**Summary:**

The paper presents an approach to combine topological features to time-series forecasting.
Essentially, the method computes 1D topological barcodes at each time-steps, pass them to a transformer and then provides those features as additional input to NBEATS. Experiments are conducted on two datasest when performing local forecast (e.g. forecasting the time-series independently) and on M4 when using global forecast (learning the model jointly across time-series). An improvement of 0.4%/1.2% is reported on M4 for OWA/sMAPE.


**Limitations And Societal Impact:**

Yes.

**Main Review:**

The paper is clearly written and well positioned compared to existing work both in forecasting and application of persistent homology for ML. The method is original and while reusing existing techniques from persistent homology, the contribution on time-series is novel and interesting on its own.

My main concern for this paper lies on the experiments. Tiny improvements are reported compared to existing approach and it was not clear how much improvement can be attributed to the technique proposed compared to other factors (noise, HPO).

First, the paper reports a specific setup for their method where learning rates are adjusted per components: for instance for the univariate setting, the learning rates are 9e-2 (for topological feature mapping), 5e-2 for the transformer taking topological features and 9e-2 for the output linear projection. For the M4 setting, different learning rates values are used. Introducing different learning rates is problematic, it would require an additional HPO budget (especially given that the two settings have different values) and it is not clear if the improvement reported by the authors would hold if a similar setup would have been used for NBEATS. I believe, a single learning rate should be used in all models to have a more apple-to-apple comparison.

Second, the results reported by the authors are very small (0.4%/1.2% improvement on M4 for OWA/sMAPE). This also raises a few questions. The noise on M4 is very large: different network initializations may give results differences that are close to the range reported by the author. As such, multiple seeds should be run to clearly show that the improvement reported is not simply due to noise. Also, while a useful ablation study has been carried out, it is a bit insufficient in my view. Part of the improvement could be done solely to the increase of parameters. As such, an ablation should also consider a model trained with roughly the same number of parameters to validate that the improvement comes from the topological features and could not have been achieved by "just" increase the hidden size of NBEATS.

I hope those points can be improved by the author during the rebuttal. One thing I was hopping when reading the paper was also to provide more insights or qualitative visualizations on the topological features as they are quite new for time-series/ML readers (why do they work, how do they help to detect complex seasonality pattern).

Additional remarks:

- l148 the description barcode coordinate function should be included in order to make the paper self-contained, some space can be gain in this section by just saying in words that you concatenate the e topological features at each time-steps
- l196: I agree that the computational complexity would not suffered with the computation of barcodes. However, a more important discussion should be on the number of additional parameters to the initial method.
- l249: how did you find the learning-rate per component? It would be challenging for the method to tune 3 learning-rate on each dataset and this diminish the value of the contribution.
- Table 2 what are (2) and (10)?
- l288: "However, in the latter strategy, the input dimensionality for the transformer
 encoder scales with the sliding window size n." I do not find this argument convincing, one could slice the number of observations to the same dimensionality as your topological representation.
- l324: "1.2% improvement over the M4 winner" this number has to be compared with the noise obtained with multiple seeds.
- l69: *Single* time series experiments


-- edit after rebuttal

I would like to thank the authors for the clear answer to the points raised in my reviews regarding hybrid learning-rates, difference in number of parameters and improvement reported compared to noise sensitivity (the latter point that seems has been noted by the other reviewers as well).

The answers and detailed analysis clarified my doubts, there is an improvement that can be clearly attributed to the method proposed (but require careful examination, I hope the analysis can be included in the manuscript). While the improvement is small, the benchmark considered (M4) is also highly competitive and the method is original and promising, as noted by other reviewers, as such I will raise my score.




**Time Spent Reviewing:**

3.5

---

> ### Author Response · Authors · 2021-08-09
> **Response to Questions/Comments**
>
> &#9654; **Regarding different learning rates among models/experiments**
>
> We agree that using a different learning rate setup for models in the *single time series* experiments and the *M4* experiments is suboptimal.
>
> Hence, we repeated **all** *single time series* experiments and can already report results on `car-parts`. As we focussed on the requested M4 experiments (see below) for these initial comments, we will update `electricity` once done.
>
> As suggested, we adopted the learning rates as reported on M4, but scaled by a factor of 10. Regarding the latter choice, we noticed that models trained on single time series typically require larger learning rates, most likely due to the reduced variation in the batches throughout training (as we sample from a single time series).
>
> So far, the results under the updated learning rates only change marginally. Most importantly, the ranking (in Table 1) remains unchanged. For a better overview, the updated table (for `car-parts`) is listed below. A change in numbers is indicated by '&rarr;', i.e., listing the originally reported and the updated value.
>
> **Updated Table 1**
>
> | `car-parts`  | Avg. Rank  | % Diff.    |
> |---|---|---|
> | Lin.+TopAttn    | 1.50 | 2.81  &rarr; 1.73 |
> |Prophet          | 2.75 | 4.91  &rarr; 4.05 |
> |MLP              | 3.00 | 7.49  &rarr; 5.99 |
> |DeepAR           | 3.50 | 7.86  &rarr; 8.24 |
> |autoARIMA        | 5.25 | 13.45 &rarr; 13.82 |
> |LSTM             | 6.00 | 16.24 &rarr; 16.54 |
> |MQ-RNN           | 7.50 | 34.36 &rarr; 34.65 |
> |Naive            | 7.75 | 30.08 &rarr; 30.39 |
> |MQ-CNN           | 7.75 | 29.19 &rarr; 29.49 |
>
> We also like to point out that the reported different learning rates for each component of our approach were not obtained via any type of hyperparameter optimization (HPO). We mostly stuck to standard settings as reported in prior works, e.g., 1e-3 is the default learning rate for NBEATS in [[29](https://arxiv.org/abs/1905.10437)], for the Transformer encoder we even omitted the customary practice of a *warmup phase* and the vectorization layer (TopVec) for persistence barcodes is typically trained (see Hofer et al. [[19](https://jmlr.org/papers/v20/18-358.html)]) with learning rates in [1e-2,1e-1].
>
> &#9654; **Regarding minor improvements over NBEATS**
>
> We agree that improvements are small in absolute numbers. Yet, on the large-scale M4 dataset, even a 1% improvement is notable, considering the corpus size of 100k time series. In fact, the 2nd to 5th top approaches in the M4 competition range within 1%-4.5% of the M4 winner (see [[28](https://www.sciencedirect.com/science/article/pii/S0169207019301128)]).
>
> We would also like to clarify that the reported performance numbers for NBEATS and our proposed NBEATS+TopAttn approach are based on an *ensemble* of models (see Section 4.5). This is the setup as reported in the original NBEATS work. Each ensemble is formed from 40 models trained with different random seeds (10) and varying horizons of historical data (2H,3H,4H,5H). The following comment specifically addresses the reviewer's question regarding differences across varying random initializations (seeds).
>
> &#9654; **Results over multiple random initializations**
>
> As already mentioned, our *ensembles* include models which differ only by the initialization from which they are trained, i.e., 10 initializations per training horizon (2H, 3H, 4H, 5H). To address the reviewer's concerns whether the reported improvements are due to noise, we assessed whether NBEATS and our NBEATS+TopAttn approach differ when fixing each horizon and then testing for differences in the SMAPE/MASE population means, obtained from the 10 models per horizon. To that end, we used a non-parametric Wilcoxon signed-rank test at $\alpha=0.05$ (+ Bonferroni correction to account for multiple comparisons).
>
> We found that in cases where NBEATS yielded a *lower* mean SMAPE/MASE score than our approach, the null hypothesis of equal population means **could never** be rejected. In other words, even if an NBEATS model yielded better results for a specific horizon, the difference was not significant. *On the other hand*, across all cases where our model performed better, the null hypothesis of equal population means was rejected multiple times, especially in the large categories of Yearly, Monthly, and Quarterly.
>
> We hope this provides further insight as to why we empirically observe improvements of adding topological attention to NBEATS.
>
> &#9654; **Additional ablation regarding model size**
>
> Considering the reviewer's comment on differences in the **number of parameters**: NBEATS has a total of 1.7M (1,740,690) parameters, our approach adds about 700k additional parameters (i.e., 2,453,740 in total). To address the question of whether improvements are simply due to a larger overall model, we ran additional experiments increasing the size of vanilla NBEATS (by doubling the hidden dimensionality) to 6.4M (6,429,330) parameters. Given the time constraints for these initial comments, we report results for a training horizon of 2H.
>
> First, we discuss **OWA** results for the corresponding *ensembles* consisting of 10 models trained on 2H. *NBEATS-large* achieves an OWA of 0.826 *vs.* 0.827 for NBEATS and 0.824 for NBEATS+TopAttn (the latter two numbers correspond to the results in Table 4). Hence, substantially increasing the number of parameters for NBEATS does improve the overall score (as expected), but does not reach the OWA result of our model.
>
> As per the reviewer's earlier request on detailed results for different random seeds, the table below lists the *median* over the SMAPE scores from 10 random initializations of each model.
>
> | `2H` | NBEATS-large (6.4M)  |  NBEATS (1.7M) | NBEATS + TopAttn (2.5M) |
> |---|---|---|---|
> *Daily (4227)*  |  2.989       | 2.977  | **2.975**  |
> *Weekly (359)*       |  9.341       | 9.394  | **9.274**  |
> *Monthly (48,000)*   |  **12.305**  | 12.348 | 12.330     |
> *Quarterly (24,000)* |  9.964       | 9.958  | **9.893**  |
> *Yearly (23,000)*    |  13.519      | 13.529 | **13.441** |
> *Hourly (414)*       |  **11.217**  | 11.639 | 11.512     |
>
> &#9654; **Visualizations**
>
> Thank you for this comment. In a final version, we will include (in the appendix) visualizations of the attention module in order to highlight, on some examples, which chunks of historical observations contained relevant *topological features* for the forecasting model.
>
> &#9654; **Addressing additional remarks**
>
> We thank the reviewer for the careful read, and we will address the raised issues in a final version. **In particular**,
> - including a full description of the specific barcode vectorization scheme early in the paper (in Section 3, as mentioned above),
> - reporting the number of parameters of the models (including results for the NBEATS-large model, as described above),
> - and fixing typos.
> - *ad Table 2*: (4) and (10) refer to the number of time series in the single time series experiments, i.e., 4 for `car-parts` and 10 for `electricity`. We will make this more clear from the table.
> - *ad l288*: It is true that one could slice the number of observations into *smaller* sliding windows with the same size as the output of the barcode vectorization. However, in that case, local context would be reduced if the resulting sliding windows are of insufficient length. Also, the number of sliding windows would increase and thereby increase the computational overhead for attention. In our setting, the sliding window size has no effect on the dimensionality of the (vectorized) topological summary. While one could use MLPs or 1D convolutions to reduce the dimensionality/length of the sliding windows of raw observations, we did not explore this.

---

> > ### Comment · Reviewer_Erx9 · 2021-08-12
> > **Answer to rebuttal.**
> >
> > Thank you for the detailed answer, I updated my score (see the edit to the review).
> >
> > Releasing the code would be quite important given the proximity of results, would you be releasing it? (the code attached contains only the block but the evaluation glue is probably critical given the amount of operations that are needed).

---

> > > ### Author Response · Authors · 2021-08-13
> > > **Source code & Inclusion of additional experiments**
> > >
> > > *Thank you very much for the quick response to our replies - we do appreciate it.*
> > >
> > > **Inclusion of experiments**: As the requested supplemental experiments and the dissection of the current results provide additional insight and allow to better isolate the reported performance gains, *we will include* them in the manuscript (following your suggestion; possibly in an extended ablation section in the appendix for space reasons).
> > >
> > > **Source code**: Yes, we will be publicly releasing the *full* source code on GitHub. We are currently in the process of preparing the public release. This will also include all the evaluation code.
> > >
> > > Fortunately, M4 provides a very clear evaluation protocol (measures, train/test split,  etc.) and, in our experiments, we exactly follow this protocol.
> > >
> > > Nevertheless, if requested, we can also provide access to all evaluation code during the review period, e.g., via an anonymous repository.

---

> > > > ### Comment · Reviewer_Erx9 · 2021-08-13
> > > > **Source code & Inclusion of additional experiments**
> > > >
> > > > Thanks for your answer and valuable additional experiments.
> > > >
> > > > It is great to know that you will contribute the full evaluation code which will be valuable.
> > > >
> > > > No need to add it right now, this comment should be binding on open-review ;-)

---

> > > > > ### Author Response · Authors · 2021-08-19
> > > > > **Update**
> > > > >
> > > > > For completeness, and as mentioned/promised in our original response, we report the updated `electricity` result of Table 1 below. *Similar to `car-parts`, the ranking remains stable*.
> > > > >
> > > > > | `electricity` | Avg. Rank  | % Diff.    |
> > > > > |---|---|---|
> > > > > | Lin.+TopAttn  | 1.50  &rarr; 1.60 |  10.59 &rarr; 5.43 |
> > > > > | DeepAR        | 1.90  &rarr; 1.80 |  12.14 &rarr; 7.71 |
> > > > > | MLP           | 2.90              |  15.39 &rarr;14.54 |
> > > > > | MQ-CNN        | 4.70              |  45.44 &rarr;45.07 |
> > > > > | autoARIMA     | 5.10              |  45.67 &rarr;45.31 |
> > > > > | Prophet       | 6.10              |  61.55 &rarr;61.28 |
> > > > > | LSTM          | 7.40              |  69.89 &rarr;69.70 |
> > > > > | MQ-RNN        | 7.50              |  68.74 &rarr;68.53 |
> > > > > | Naive         | 7.90              |  77.71 &rarr;69.82 |

---

### Official Review · Reviewer_DXFs · 2021-07-17

**Rating:** 6
**Confidence:** 3

**Summary:**

The paper proposes topological attention, a type of self-attention mechanism that captures the topological information of the signals. In contrast to earlier works, this work focuses on local instead of global topological summaries. The main difference is attributed to the extraction of such summaries for each sliding window and not over the global historic data. The paper discusses how such sets of local summaries can be vectorized and incorporated into neural network architectures. The experimental results on an existing forecasting approach show the benefit of topological attention vs. the regular (global) attention mechanism.

**Limitations And Societal Impact:**

Yes

**Main Review:**

The paper is well-written and the introductory figures help in the understanding of the proposed approach.

The novelty of the paper is somewhat limited, considering that, conceptually, the main difference is the use of an existing approach over sliding windows. However, the vectorization and integration of such extracted topological summaries are not trivial and, therefore, there is merit in having such an approach documented.

The results are not impressive. For the single time series experiments the method used to incorporate the topological attention seems to already perform surprisingly well due to the simplicity of the task. However, it is interesting to see that standalone topological features without attention mechanism reduce performance and, therefore, the benefit of using both approaches combined becomes evident.

The results on the M4 forecasting competition show similarly not impressive trends: the performance depends on the horizon, some times a simple attention mechanism performs better. However, even a 1% improvement might be statistically significant considering the size of the corpus here. The ablation study creates a much clearer picture of the benefits.

**Time Spent Reviewing:**

3

---

> ### Author Response · Authors · 2021-08-09
> **Response to Questions/Comments**
>
> &#9654; **Regarding the novelty of our approach**
>
> We believe the novelty of this work is rooted in the **departure** from the common approach (adopted by earlier works) for topologically analyzing time series data: specifically, we depart from converting a *large* segment of observations into a point cloud and studying Vietoris-Rips persistent homology. Such an approach inevitably loses *local* information, as one would obtain ONE persistence barcode, encoding topological features of the full segment - we argue that this suboptimal in forecasting problems, as the relevance of locality has already been highlighted in recent works (see [[24]](https://papers.nips.cc/paper/2019/file/6775a0635c302542da2c32aa19d86be0-Paper.pdf)).
>
> To address the *locality* challenge, we (1) compute topological features from a sub levelset filtration of *shorter* segments (i.e., the sliding windows) of *raw* observations and (2) then letting the model learn to attend to segments that are relevant to the forecasting task. To the best of our knowledge, this approach of incorporating topological features into a forecasting pipeline, especially in large-scale settings, has not been explored so far. However, we do agree that each building block of our approach (when considered on its own) does already exist.
>
> &#9654; **Regarding "non-impressive" results**
>
> We would like to clarify that on the large-scale M4 dataset, attention *without* topological features *does not* perform better than attention *with* topological features, see Table 4. The same is true for *single time series* experiments on `electricity`. On the small-scale *single time series* experiments for `car-parts`, the reviewer's remark is correct in that directly feeding observations from sliding windows into the attention module already performs surprisingly well.
>
> As remarked by the reviewer, considering the corpus size of 100k time series on M4, even a 1% improvement is notable. In fact, the 2nd to 5th top approaches in the M4 competition range within 1%-4.5% of the M4 winner (see [[Table 4, 28](https://www.sciencedirect.com/science/article/pii/S0169207019301128)]).

---

### Decision · Program_Chairs · 2021-09-27

**Decision:**

Accept (Poster)

**Comment:**

The paper proposes a form of local topological attention to improve the performance of a time series forecasting model. The reviewers agree that the proposed method is interesting and relevant, and the writing is clear.
On the other hand, all reviewers caution that the performance improvements demonstrated empirically are small (~ 1% on M4), and two reviewers see the novelty as somewhat limited. However, the authors were able to alleviate further concerns regarding the empirical evaluation during the discussion period, leading all reviewers to vote for (weak) acceptance. The reviewers belief that the merits outweigh the flaws of the paper, and I concur with their assessment, recommending acceptance.